# Gossypol and Its Natural Derivatives: Multitargeted Phytochemicals as Potential Drug Candidates for Oncologic Diseases

**DOI:** 10.3390/pharmaceutics14122624

**Published:** 2022-11-28

**Authors:** Dilipkumar Pal, Pooja Sahu, Gautam Sethi, Carly E. Wallace, Anupam Bishayee

**Affiliations:** 1Department of Pharmaceutical Sciences, Guru Ghasidas Vishwavidyalaya (A Central University), Bilaspur 495 009, India; 2Department of Pharmacology, Yong Loo Lin School of Medicine, National University of Singapore, Singapore 117600, Singapore; 3College of Osteopathic Medicine, Lake Erie College of Osteopathic Medicine, Bradenton, FL 34211, USA

**Keywords:** gossypol, antitumor effects, apoptosis, autophagy, in vitro, in vivo, drug development

## Abstract

Despite the vast amounts of research and remarkable discoveries that have been made in recent decades, cancer remains a leading cause of death and a major public health concern worldwide. Gossypol, a natural polyphenolic compound derived from the seeds, roots, and stems of cotton (*Gossypium hirsutum* L.), was first used as a male contraceptive agent. Due to its diverse biological properties, including antifertility, antiviral, antioxidant, antibacterial, antimalarial, and most notably antitumor activities, gossypol has been the subject of numerous studies. Nevertheless, no systematic review has been performed that analyzes the antineoplastic potential of gossypol and related natural compounds in an organ-specific manner while delineating the molecular mechanisms of action. Hence, we have performed an extensive literature search for anticancer properties of gossypol and their natural derivatives against various types of cancer cells utilizing PubMed, ScienceDirect, Google Scholar, and Scopus. The sources, distribution, chemical structure, and toxicity of gossypol and its constituents are briefly reviewed. Based on emerging evidence, gossypol and related compounds exhibit significant antineoplastic effects against various cancer types through the modulation of different cancer hallmarks and signaling pathways. Additionally, the synergistic activity of gossypol and its derivatives with chemotherapeutic agents has been observed. Our evaluation of the current literature suggests the potential of gossypol and its derivatives as multitargeting drug candidates to combat multiple human malignancies.

## 1. Introduction

Gossypol is a lipid-soluble polyphenol that is isolated from the cotton plant (genus Gossypium) and the tropical tree *Thespesia populnea* (L.) Sol. ex Corre (family Malvaceae). The chemical name for gossypol is (2,2′-binaphthalene)-8,8′-dicarboxaldehyde,1,1′,6,6′,7,7′-hexahydroxy-5,5′-diisopropyl-3,3′-dimethyl), which is the most important natural pigment present in cotton (*Gossypium hirsutum* L.) [1]. Longmore and Marchlewski isolated and crystallized gossypol at the end of the 19th century using the name “gossypol” to reference its origin from the Gossypium genus and phenolic character [2,3]. Gossypol (molecular formula: C_30_H_30_O_8_) is high in fiber and low in protein content (Figure 1) [4]. The seeds of *G. hirsutum* produce gossypols in their free and bound form [5,6]. Both types are created during the growth and maturation stages of development, although the ratios may fluctuate when stored or depending on how the cottonseed oil is extracted [7,8]. The free form of gossypol is toxic in nature, causing detrimental consequences, such as antisterility in animals and humans, whereas the bound form is harmless when attached to proteins [9,10,11]. By interacting with the epsilon amino group of lysine, an important amino acid, bound gossypol decreases the nutritional value of protein, the bioavailability of lysine, and the digestibility of cottonseed meal [12,13,14,15]. While gossypol was first used as a male contraceptive [16], further research has shown that this polyphenol has both nutritional and medicinal properties, including antiviral, antioxidant, antiparasitic, and antibacterial activities [9]. More recently, gossypol has been investigated for its anticancer activity against various cancers, including breast, colon, pancreatic, and prostate cancers [15,16]. There have been at least two previous reviews of gossypol, each of which have their own limitations. Keshmiri-Neghab et al. [17] published a short review of the chemical and biological properties of gossypol. While antitumor properties were briefly discussed, this was not the primary focus of the article. In a more recent review by Zeng et al. [18], the anticancer mechanisms of gossypol and its derivatives were summarized based on in vitro and in vivo studies. However, gossypol has been investigated for its anticancer activity.

During the last several decades, numerous studies have reported promising anticancer properties of gossypol analogues in both preclinical (i.e., in vitro and in vivo) and clinical studies. Although many studies have investigated the anticancer effects of gossypol, no systematic review has been performed that summarizes these findings in an organ-specific manner while delineating the molecular mechanisms of action. In view of this limitation, this comprehensive study critically evaluates the antineoplastic efficacy of gossypol and its natural and synthetic derivatives. The intent for this review is to describe recent advances in the chemotherapeutic potential of gossypol and related compounds against various malignancies and in combination with other chemotherapeutic agents, identify limitations in the current literature, and suggest future research directions. Emphasis is placed on describing the molecular pathways through which gossypol and its natural analogs exert anticancer activities, illuminating their potential as multitargeting drug candidates to combat multiple human malignancies.

## 2. Sources and Distribution of Gossypol

Gossypol was discovered in 1886 by Longmore, which was later refined in 1899 by Marchlewski when gossypol acetic acid was precipitated from an ether solution with acetic acid. Because of its origin in the genus Gossypium and its polyphenolic chemical composition, the product was termed gossypol [3,4]. Throughout the 1950s, the antifertility effect of gossypol in mammals and humans was validated in multiple studies [5,16]. The dimerization of two molecules of hemigossypol yields the dimeric-sesquiterpenoid gossypol. Sesquiterpenoids are terpenes with three isoprene units that protect plants from infections and insects [17,19].

Gossypol is a bright yellow pigment produced in the intracellular pigment glands of cotton stems, leaves, seeds, laproot bark, boll valves, seed hulls, pericarp, and flowers (Table 1) [9,20,21]. On the sliced surface of cottonseed, these pigment glands appear as small black spots. Factors that influence the quantity of gossypol includes the amount of fertilizer used, the composition of the fertilizer, the species of cotton plant, sowing times, regional temperatures, soil conditions, water distribution, and agrotechnical treatment [6,20,22].

## 3. Chemistry of Gossypol: Structure, Atropisomerism, Stereochemistry, and Tautomerism

Apogossypolone, apogossypol, gossypolone, 6-aminopenicillanic acid sodium gossypolone, and BI-97C1 are all derived from gossypol (Figure 2A,B) [23,24]. Various gossypol derivatives and their structures are summarized in Table 2. Gossypol is a racemic combination of polyphenolic bissesquiterpenes isolated from cottonseed. Due to the hindered rotation around the binaphthyl link, gossypol exists in two enantiomers: (+) and (−) [24,25,26]. Racemization of gossypol requires an incomprehensibly high amount of energy; hence the individual enantiomers are optically stable under typical conditions (e.g., ambient temperature and neutral pH) [27,28,29].

Three tautomeric forms of gossypol are described in order to differentiate their unique reactions, characteristics, and degradation products. These tautomeric forms include aldehyde, ketone (quinoid), and lactol (hemiacetal). (−)-Gossypol has shown greater potency than (+)-gossypol or racemic gossypol in several biological evaluation experiments [6,30].

### 3.1. Atropisomerism

Gossypol introduces a type of enantiomerism called atropisomerism. Their two optically active forms are l- or (−), (−)-1, (−)-gossypol and d- or (+), (+)-1, (+)-gossypol that occur from the restricted rotation about the C_2_–C_2_′ dinaphthalene bond (Figure 2C,D) [31]. The absolute configuration of the (−)-enantiomers have been named (aR)-gossypol (M), while (+)-enantiomers are called (aS)-gossypol (P), where (M) and (P) denote the two helical forms of gossypol that exist because of the restricted internaphthyl bond rotation [25].

### 3.2. Tautomerism

Gossypol exhibits two symmetrical and one asymmetrical tautomeric form called aldehyde, ketol, and lactol. The ketol tautomer is present in aqueous alkaline solutions, but is not stable in neutral and acidic solutions, where it is quickly converts to the aldehyde form [6,32]. An equilibrium occurs between the aldehyde and lactol tautomer in polar solvents, which favors the lactol form when the nucleophilicity of the solvent rises [9,33]. The presence of metal ions also impacts the activity of these tautomeric forms. The symmetrical ketol tautomer binds with iron (Fe^2+^ or Fe^3+^) and zinc (Zn^2+^) cations to create stable complexes, whereas nickel (Ni^2+^) and copper (Cu^2+^) cations shift the tautomeric equilibrium towards the lactol tautomeric form (Figure 3) [6,9,34].

## 4. Toxicity of Gossypol

Gossypol can be utilized as a male contraceptive, as well as for the treatment of cancer and microbiological diseases [17,35]. The effect of gossypol on normal genetic processes must be understood before it can be regarded as safe for use in humans, in particular healthy women of reproductive age. Although various laboratory and clinical studies have addressed concerns relating to gossypol’s genetic consequences, no comprehensive method of characterizing its genotoxic potential has been performed. Most of the positive effects observed either (1) are likely to be eliminated or insignificant in vivo at expected clinical doses when in the presence of normal serum protein levels, or (2) can be explained by mechanisms involving changes in enzymes and other cell components involved in DNA replication rather than direct interactions with DNA. Nonetheless, given the potential risks of employing a drug as a contraceptive agent before the genetic side effects are fully established, more research into the direct effects of gossypol and its mechanisms of action in normal versus tumor cells is crucial. Not only would this information be useful in determining the safety of gossypol, but also in identifying potential new uses for the substance [36,37,38,39]. Given the proapoptotic characteristics and synergism with other anticancer treatments, gossypol displays antitumor potential. However, due to its poor water solubility and gastrointestinal toxicity, its applications may be limited. Patients receiving oral gossypol treatment for breast cancer experienced nausea, vomiting, and exhaustion. Many gossypol derivatives have been developed to improve their shortcomings, such as weak antitumor activity, considerable side effects, poor water solubility, and lack of cancer cell spatial specificity [40].

## 5. Pharmacokinetic Profile of Gossypol

Apogossypol and gossypol exhibit similar stability in plasma from various species, despite 20–40% of apogossypol hexaacetate transform into apogossypol as well as di-, tri-, tetra-, and penta-acetates of apogossypol. Mice treated with oral and intravenous (i.v.) (±)-gossypol and (−)-gossypol demonstrated comparable pharmacokinetic profile and oral bioavailability (12.2–17.6%), with some changes in clearance and volume of distribution at steady state [1]. Apogossypol demonstrated delayed time to reach maximum concentration (Tmax), a slower clearance rate, and poor distribution after injection to mice at the same molar dose. Following injection, apogossypol mono- and di-glucuronide conjugates were easily detected in mouse plasma. The area under the curve (AUC) and oral bioavailability of apogossypol in sesame oil was greater than that of cremophor:ethanol saline. In contrast, the maximum clearance rate was observed with i.v. apogossypol hexaacetate, perhaps due to its conversion to apogossypol. Agossypol produced from apogossypol hexaacetate was quantitatively discovered concurrent with the elimination of i.v. apogossypol hexaacetate, and accounted for about 30% of the total plasma apogossypol hexaacetate. Minimal apogossypol was found in the plasma after oral apogossypol hexaacetate administration, indicating poor bioavailability. Except for gossypol glucuronidation, glucuronide conjugates of apogossypol and its acetates were easily recognized in human and mouse liver microsomes [28]. Gossypol and apogossypol hexaacetate are less stable in human and mouse liver microsomal preparations than apogossypol [11].

## 6. Methodology for Literature Search and Selection of Anticancer Studies

The Preferred Reporting Items for Systematic Reviews and Meta-Analysis (PRISMA) criteria was used for this study, which is the recommended method for conducting systematic reviews [41]. An extensive literature search was conducted utilizing various databases, such as PubMed, ScienceDirect, Google Scholar, and Scopus. Major keywords used in multiple combinations included: Gossypol, cancer, anticancer, tumor, antitumor, carcinoma, apoptosis, autophagy, Bcl-2, reactive oxygen species, in vitro, in vivo, clinical study, and drug development. The title and abstract of all publications were initially assessed for their relevance to gossypol and cancer before full-length articles were retrieved and examined. A mutual decision was made if there was any uncertainty whether an article should be included for further analysis. Only articles written in the English language were evaluated. Review articles, meta-analyses, letters to the editors, book chapters, abstracts, and unpublished results were excluded. Clinical trials pertaining to gossypol and cancer were checked on clinicaltrials.gov. The reference lists of previous reviews and collected articles were examined for relevant publications. A total of 48 articles met the selection criteria. A summary of the literature search and study selection is shown in Figure 4.

## 7. Anticancer Activities of Gossypol

### 7.1. Adrenal Cancer

Adrenocortical carcinoma (ACC) is a devastating malignancy for both patients and their relatives due to the short life expectancy and significant impact on quality of life, which is impacted by the presence of metastasis, accompanying endocrine disorders, and treatment-related side effects. All adrenal masses are thoroughly investigated to rule out neoplasm, because although the majority of adrenocortical tumors are benign, it is crucial to identify the rare instances of ACC. Clinical suspicion for ACC includes symptoms of hormone excess, which typically causes Cushingoid features as a result of excess glucocorticoids, but can also present as virilization due to androgen overproduction. In non-functioning neoplasms, symptoms occur due to tumor bulk, most often causing abdominal pain. An incidental adrenal mass may also be observed on imaging [44,45].

Leblanc et al. [46] investigated the anticancer effects of gossypol and apogossypol hexaacetate against SW-13 and H295r ACC cell lines. Both gossypol and apogossypol hexaacetate inhibited cell proliferation (Table 3). However, no mechanism of action was proposed.

### 7.2. Bladder Cancer

Bladder cancer is the most common urinary system malignancy and is associated with a high rate of morbidity and mortality. Bladder cancer is a heterogeneous disease, with 70% of patients presenting with surface tumors that recur but are not life threatening, while 30% of cases involve invasion into the muscle, which carries a greater risk of distant metastases. Painless hematuria is the most common presenting symptom. Diagnostic workup includes urine cytology and cystoscopy, which is the gold standard for diagnosis and staging [83,84]. Targeting apoptotic pathways is a potential chemotherapeutic mechanism by altering the substrates involved in programmed cell death, namely caspases and B-cell lymphoma 2 (Bcl-2) proteins. Dysregulation of Bcl-2 proteins has been linked to a poor chemotherapy response in patients with bladder cancer [85].

The cottonseed polyphenol (−)-gossypol, a BH3 mimetic, increased apoptosis in chemosensitive UM-UC2 and chemoresistant UM-UC9 bladder cancer cells in vitro. The proposed mechanism was via decreased activation of both caspase-3 and caspase-9, with more pronounced effects in UM-UC9 cells in a concentration- and time-dependent manner. Additionally, combination of gossypol, gemcitabine, and carboplatin increased apoptosis in UM-UC9 cells only through downregulation of B-cell lymphoma-extra-large (Bcl-xL) and myeloid cell leukemia 1 (Mcl-1) and overexpression of Bcl-2-like protein 11 (Bim) and p53 upregulated modulator of apoptosis (PUMA). The prosurvival proteins Mcl-1 and Bcl-xL sequester other pro-apoptotic proteins, such as Bak, inhibiting their activation. Therefore, removing Mcl-1 through RNA interference of tumor cells may improve sensitivity to chemotherapy agents. However, more clinically amenable methods must be found to reduce Mcl-1 expression [47].

The prosurvival Bcl-2 family of proteins are a key target in bladder tumors, and both natural and synthetic antiapoptotic molecules can disrupt these proteins at various stages. As a part of the aforementioned study, Macoska et al. [47] found that when both UM-UC2 and UM-UC-9 cells were treated with (−)-gossypol in vitro, there was increased cell sensitivity to BH3 mimetics, which can act as potent Bcl-2 prosurvival family inhibitors. Furthermore, adding (−)-gossypol to chemotherapy regimens may improve response rates in patients with advanced bladder cancers.

### 7.3. Breast Cancer

According to the World Health Organization, 2.3 million women were diagnosed with breast cancer in 2020. Not only does the incidence of breast cancer continue to rise, but it is now the most prevalent cancer in the world, with over 7.8 million women affected [86]. In regard to cancer-related mortality, breast cancer is the second most fatal cancer among women, with 685,000 deaths in 2020 alone. The mortality caused by breast cancer has fallen in North America and the European Union due in part to early identification and more effective systemic therapy options [87].

Breast tissue is composed of glandular and stromal (supporting) tissues. Glandular tissues contain the milk-producing glands (lobules) and ducts (milk tubes), whereas stromal tissues contain the fatty and fibrous connective tissues [88]. Breast cancer most often develops along the milk ducts or lobules [87,89]. When breast cancer metastasizes, it often spreads to the regional lymph nodes and to distant organs, such as the bones, liver, lungs, and brain, making it extremely difficult to cure. Early detection is key to achieving an excellent prognosis and a high survival rate. Mammography is the most common screening method used for detecting breast cancer and has been shown to effectively reduce mortality. Other screening modalities, such as magnetic resonance imaging, have been explored in the last decade and may be more sensitive than mammography. The risk of breast cancer is multifactorial and includes but is not limited to sex, age, length of estrogen exposure, and family history [90].

Gilbert et al. [43] established that in both hormone-dependent and hormone-independent human breast cancer cell lines (MCF-7, MCF-7Adr, and MDA-MB-231), gossypol and gossypolone, an oxidized metabolite of gossypol, decreased DNA synthesis and cell proliferation in a concentration-dependent manner in vitro. Gossypolone was found to be less effective than gossypol, despite gossypolone having antisteroidogenic and antireproductive properties. In a subsequent study, the anticancer effects of the gossypol on the triple-negative breast cancer cell lines, MDA-MB-231 and MDA-MB-468, was investigated. Gossypol induced cytotoxicity and reduced proliferation in both cell lines. MDA-MB-468 cells, on the other hand, were twice as sensitive to the compound’s apoptotic effects, which was followed by a longer delay in colony formation. Growth arrest and DNA damage-inducible 45α protein (GADD45α), tumor necrosis factor receptor superfamily 9, and BCL2 interacting protein 3 (BNIP3) were all elevated in MDA-MB-231 cells treated with gossypol, while baculoviral IAP repeat containing 5, an apoptosis-suppressor gene, and death-associated protein kinase 1 were decreased. In MDA-MB-468 cells there was increased GADD45α and BNIP3 and reduced tumor protein 73. The 159-fold increase in TNF gene expression seen in MDA-MB-468 cells is a novel discovery [49]. Ye et al. [52] observed that combination of (–)-gossypol-enriched cottonseed oil ((–)-GPCSO) with the chemotherapeutic drugs tamoxifen, ICI 182 780, or adriamycin (Adr) suppressed cell growth of the multidrug resistant human breast cancer cell line (MCF-7/Adr) and primary cultured human breast cancer epithelial cells (PCHBCEC) through decreased multidrug resistance protein expression in vitro.

### 7.4. Colorectal Cancer

Colorectal cancer is the third most commonly diagnosed cancer worldwide as well as the third leading cause of cancer death. Colon cancer is more common than rectal cancer in industrialized countries, with a 2:1 ratio of colon to rectal cancer. In comparison, in non-industrialized countries the rates of colon and rectal cancer are generally equal. More than 250,000 new cases of colon cancer are diagnosed annually in Europe, accounting for around 9% of all cancers. Historically, colorectal cancer was much more prevalent in high-income countries. However, with increasing industrialization and urbanization, the incidence of colon cancer continues to rise and is becoming more prevalent in middle- and low-income countries [91].

The cotton-derived polyphenolic substance gossypol has demonstrated significant anticancer properties against colon cancer. In the human colon cancer cell line COLO 225, gossypol reduced cell viability and suppressed the expression of claudin-1, Elk-1, fatty acid synthase, glyceraldehyde 3-phosphate dehydrogenase, interleukin (IL)-2, IL-8, and zinc finger, AN1-type domain 5, while glucose transporter 3 mRNA was increased. Gossypol decreased the ability of the cells to form colonies and reduced cellular internucleosomal DNA fragmentation, which accompanied by the emergence of a subG1 apoptotic peak, supported that gossypol induced cell death through an apoptotic route, which may be cell cycle independent. Furthermore, gossypol induced apoptosis through decreased expression of Bcl-2 and Bcl-2-associated X protein (Bax) [8].

Zhang et al. [54] treated HT-29 human colorectal cancer cells with gossypol, which increased apoptosis through improved activation of caspase-3, caspase-6, caspase-7, caspase-8, and caspase-9. In another study, (−)–gossypol induced mitochondrial-mediated apoptosis and enhanced release of reactive oxygen species (ROS) in HCT116 human colorectal cancer cells through increased susceptibility to tumor necrosis factor related apoptosis-inducing ligand [64]. In conclusion, gossypol may be useful as an adjuvant treatment for colorectal cancer. Further research regarding the specific anticancer mechanisms through which gossypol acts is critical for its implementation into human clinical trials.

### 7.5. Gastric Cancer

Gastric cancer is prevalent in Eastern Asian countries, affecting males more often than females. The majority of gastric cancer cases occur in resource-limited countries and is a significant cause of mortality. Gastric cancers can be categorized depending on their anatomic location, including the gastroesophageal junction, proximal stomach, and distal stomach, which consists of the body and antrum. *Helicobacter pylori* is the most common cause of distal gastric cancer, although other factors that impact the development of gastric cancer include host immune response, epithelial response, and environmental factors [92,93].

The most effective gossypol derivative, apogossypolone (ApoG2), has been identified as a new small-molecule inhibitor of the Bcl-2 family proteins. In a study, ApoG2 inhibited cell growth in MKN28, MKN45, and AGS human gastric cancer cell lines in a time- and concentration-dependent manner. ApoG2 suppressed the development and proliferation of gastric cancer cells through downregulation of Bcl-2 expression, upregulation of Bax, and increased activation of caspase-3. While ApoG2 was not as effective as (−)-gossypol in reducing cell growth, ApoG2 had a higher colony formation potential. Therefore, ApoG2 may be an effective compound for suppressing the growth and proliferation of gastric cancer cells through mitochondrial-mediated apoptosis [56].

### 7.6. Head and Neck Cancers

Multiple anatomic sites in the head and neck region give rise to squamous cell carcinoma. Tobacco use and alcohol abuse are risk factors for malignancies of the oral cavity, oropharynx, hypopharynx, and larynx. Infection with oncogenic viruses is associated with cancers of the nasopharynx, palatine, and lingual tonsils. In developed countries, the annual incidence of human papillomavirus-associated oropharyngeal cancer is rising. Chemotherapy resistance is a major complication in the treatment of head and neck squamous cell carcinoma (HNSCC). Antiapoptotic proteins, such as Bcl-xL, are typically overexpressed in chemoresistant HNSCC tumors [60,61].

In an in vitro study by Wolter et al. [60], the negative enantiomer of a cottonseed polyphenol, (−)-gossypol, reduced cell proliferation in Um-SCC-1 and Um-SCC-17b HNSCC cells through binding to Bcl-xL. (−)-Gossypol also inhibited proliferation of the cisplatin-resistant cell lines UM-SCC-5PT and UM-SCC-10BPT. In the same study, orthotopic xenograft mice bearing UM-SCC-5 and UM-SCC-10B cells demonstrated reduced tumor growth in vivo when treated with (−)-gossypol.

### 7.7. Lung Cancer

Lung cancer is the leading cause of cancer death worldwide, accounting for 12.3% of all malignancies [94]. Tobacco use is the strongest risk factor for the development of lung cancer; however, many other environmental factors also contribute, including air pollution, asbestos, radon, and smoke from cooking and heating. There are significant geographic, racial, and sexual disparities in lung cancer incidence. Some studies suggest that women may be at higher risk of lung cancer due to tobacco smoke toxin exposure. Non-small cell lung cancer (NSCLC) accounts for 85–90% of lung malignancies and has been challenging to treat due to the unclear pathophysiology. Traditional treatment options for NSCLC include surgical procedures, radiation, and chemotherapy. The overexpression of epidermal growth factor receptor (EGFR) has been linked to the development of NSCLC. Molecular-targeted therapy with tyrosine kinase-based inhibitors can be utilized therapeutically in NSCLC cases when EGFR mutations are present [94]. Several EGFR inhibitors have been tested in clinical trials, however the discovery of the EGFRL858R/T790M resistant mutation has diminished the efficacy of these drugs. Novel therapeutic options are urgently needed for the treatment of EGFRL858R/T790M resistant NSCLC.

Wang et al. [72] investigated the effects of gossypol on the EGFRL858R/T790M mutated NSCLC H1975 cell line in vitro. Gossypol suppressed cell growth and migration, as well as increased caspase-dependent cell death through upregulation of Bcl-2 antagonist of cell death. Gossypol binded to the kinase domain of EGFRL858R/T790M with a high affinity through hydrogen bonding and hydrophobic contacts, thus reducing the activity of EGFRL858R/T790 kinase M. Furthermore, gossypol suppressed phosphorylation of EGFR and its downstream signal pathways in a concentration-dependent manner. These findings lay the groundwork for the development of novel EGFRL858R/T790M inhibitors for the treatment of NSCLC.

### 7.8. Multiple Myeloma

Multiple myeloma (MM) is a clonal proliferation of malignant plasma cells. Growing data suggests that the microenvironment of tumor cells in the bone marrow plays a key role in the pathogenesis [95]. According to a multistep development model, monoclonal gammopathy with unknown clinical relevance can advance to smoldering multiple myeloma, symptomatic intramedullary and extramedullary multiple myeloma, or plasma cell leukemia [96]. Oncogenomic studies have demonstrated that there are few distinctions between monoclonal gammopathy of unknown significance and multiple myeloma, highlighting the critical involvement of the bone marrow microenvironment in the genesis, maintenance, and progression of the disease [97].

Cell surface receptors, such as integrins, cadherins, selectins, syndecans, and the immunoglobulin superfamily of cell adhesion molecules, facilitate direct contact between multiple myeloma cells and bone marrow stromal cells, or between extracellular matrix proteins of multiple myeloma cells. Both forms of interactions promote multiple myeloma cell proliferation, survival, migration, and treatment resistance, as well as influence the role of bone marrow stromal cells through enhanced cytokine secretion [98].

Zhang et al. [66] found that gossypol upregulated 273 genes and downregulated 259 genes in the MM cell lines U266, MM1-144, OPM2, ARP-1, OCI-MY5, CAG, H929, KMS11, and ARK in a concentration- and time-dependent manner in vitro. In comparison to normal plasma cells, the expression of the JUN protein product c-Jun is downregulated in MM cell lines and in patients with high-risk MM. Patients with a high expression of JUN have improved mortality rates. When JUN is overexpressed in MM cells, there is increased cell death and growth inhibition through a caspase-dependent apoptotic pathway. Death receptor 5 (DR5) is an upstream receptor of the Jun N-terminal kinase (JNK) pathway that, when knocked out with short hairpin RNA, can partially restore gossypol-induced apoptosis.

In patients with MM, 1017 genes have been found to be co-expressed with JUN. Other JNK-related signaling networks, including IL-6, EGFR, and platelet-derived growth factor pathways, are predominantly regulated by these genes. In summary, gossypol can trigger caspase-dependent apoptosis through the JNK pathway by targeting c-Jun and other JNK-associated pathways. JUN has been identified as the hub gene in gossypol-induced apoptosis, involving both DR5 and IL-6.

### 7.9. Prostate Cancer

Prostate cancer is the second most common cancer in men worldwide, according to the World Health Organization, affecting one in eight men. Androgens, such as testosterone and dihydrotestosterone, stimulate the growth of prostate cancer and play a role in its pathophysiology. Prostate carcinogenesis are thought to comprise a multistep transition from precancerous cells to proliferative and metastatic cells. The growth and development of prostate cancer cells appears to initially be androgen-dependent, through transcription regulation of downstream genes via androgen receptors [99,100,101,102].

An in vitro study by Jiang et al. [77] found that racemic gossypol, (±)-gossypol, inhibited cell proliferation of the Dunning human prostate cancer cell line, MAT-LyLu. In MAT-LyLu cells, gossypol reduced the expression of cyclin D1, cyclin-dependent kinase 4 (CDK4), and phospho-retinoblastoma (p-Rb), which regulate the progression of the cell cycle in prostate cancer cells. Thus, the modulation of transforming growth factor 1 (TGF-1) and Akt signaling, which affect the expression of regulatory proteins, is linked to the inhibitory effects of gossypol on the proliferation of MAT-LyLu cells.

A thymidine incorporation assay and doubling time (DT) measurement was used to investigate the effects of (±)-gossypol on the proliferation of human prostate cancer PC-3 cells. (±)-Gossypol decreased DNA synthesis and cyclin D1 mRNA, prolonged DTs, and increased TGF-β1 mRNA expression in PC-3 cells. Adding 25 g/mL of mouse monoclonal TGF-1, TGF-2, and TGF-3 antibodies to conditioned media collected from (±)-gossypol-treated PC-3 cells, the growth inhibition of PC-3 cells was completely reversed, implying that TGF-1 is involved in (±)-gossypol induced growth inhibition of PC-3 cells. Furthermore, TGF-1 controlled the expression of the cell cycle-regulatory protein cyclin D1 [103].

The antiapoptotic proteins Bcl-2 and Bcl-xL are often overexpressed in human prostate malignancies, rendering them resistant to radiation therapy. (−)-Gossypol has recently been discovered to be a powerful Bcl-2 and Bcl-xL inhibitor, which may increase the response of prostate cancer to radiation by potentiating radiation-induced apoptosis. In a study by Xu et al. [78], radiation-induced apoptosis and growth suppression were improved by (−)-gossypol in PC-3 cells containing a high amount of Bcl-2/Bcl-xL proteins. (−)-Gossypol induced apoptosis through increased poly (ADP-ribose) polymerase (PARP) cleavage, caspase-3, caspase-8, and caspase-9 activation, and caspase-activated deoxyribonuclease (CAD) proteins, while reducing inhibitor of CAD (ICAD) proteins. By upregulating the expression and secretion of TGF- β1 and downregulating the expression of Akt and p-Akt protein, (−)-gossypol suppressed cell proliferation and colony formation in a concentration-dependent manner. (−)-Gossypol inhibited cell proliferation, as evident by cell cycle arrest in the G0/G1 phase. In a subsequent study by Huang et al. [76], gossypol increased non-metastatic protein 23 and decreased Bcl-2 and Bcl-Xl in MAT-LyLu rat prostate cancer cells. The antiproliferative effects of gossypol involves TGF- β1 and Akt signaling, which alter the expression of the regulatory proteins, resulting in reduced expression of cyclin D1, CDK4, and p-Rb.

Loberg et al. [104] studied the anticancer activity of the pan-small molecule inhibitor of Bcl-2, AT-101(R-(–)-gossypol acetic acid, against VCaP prostate cancer cells in vivo. SCID mice xenografted with VCaP cells demonstrated tumor growth inhibition through increased apoptosis, Bcl-2, and androgen receptor expression when treated with AT-101(R-(–)-gossypol acetic acid following surgical castration (Table 4). In another study, (−)-gossypol activated caspase-3, caspase-8, and caspase-9 and increased PARP cleavage in in DU-145 prostate cancer cells. Furthermore, the pan-caspase inhibitor z-VAD fmk prevented gossypol-induced apoptosis through increased CAD and decreased ICAD, as presented in Figure 5 and Figure 6 [79].

## 8. Anticancer Activities of Gossypol Derivatives

### 8.1. Apogossypol and ApoG2

ApoG2 is a derivative of gossypol synthesized through detachment of two aldehyde groups. In mice, the maximum tolerated dose of ApoG2 is two- to four-times higher than that of gossypol, with an improved side effect profile, including decreased hepatotoxicity and gastrointestinal toxicity. ApoG2 is more sensitive than gossypol to Bcl-2 gene converting B cells in mice. The stereoisomer of ApoG2, (±)-ApoG2, is theoretically comparable to (±)-gossypol; however, due to technical constraints, this derivative has yet to be isolated. As a result, the antitumor properties of ApoG2 have only been demonstrated in (±)-ApoG2 [107].

Xin et al. [56] found that human gastric cancer cell lines, MKN28, MKN45, and AGS, treated with ApoG2 demonstrated less cytotoxicity and did not as effectively impact cell survival to the extent that gossypol did in vitro. However, ApoG2 reduced colony formation to a greater extent than gossypol and inhibited gastric cancer cell growth and proliferation by downregulating Bcl-2, upregulating Bax, and activating caspase-3.

In the nasopharyngeal cancer cell lines, C666-1, CNE-1, CNE-2, and HONE-1, ApoG2 inhibited c-Myc expression through increased p21 and inactivation of cyclin D1 and cyclin E. Additionally, ApoG2 promoted cell cycle arrest in the S phase through inhibition of c-Myc and its downstream regulators. In C666-1, CNE-1 and CNE-2 cells with high Bcl-2 expression, ApoG2 induced DNA fragmentation. However, in HONE-1 cells with a low Bcl-2 level, this behavior was not observed. The maximum activities of caspase-3 and caspase-9 appeared 12 and 24 h after ApoG2 treatment, respectively [63]. A study by Hu et al. [62] investigated the effects of ApoG2 against the nasopharyngeal cell lines C666-1, CNE-1 and CNE-2 both in vitro and in vivo. ApoG2 inhibited cell proliferation and c-Myc expression, while inhibiting angiogenesis through decreased CD31 in vitro. Similar to gossypol, ApoG2 hindered heterodimer formation by blocking interactions with Bcl-2-Bax and Bcl-xL-Bak. In athymic nude (nu/nu) mice xenografted with CNE-1 or CNE-2 xenografts, ApoG2 suppressed tumor growth. ApoG2 also enhanced the anticancer effects of cisplatin (both in vitro and in vivo).

ApoG2 induced apoptosis and decreased the development and proliferation of gastric cancer SGC-7901 cells in vitro through increased endoplasmic reticulum (ER) stress, as evident by increased ROS, Ca^2+^, and GADD153, a marker of ER-stress apoptosis [57].

Zhang et al. [55] found that mice bearing PC-3 xenografted cells treated with 5 mg/kg ApoG2 had decreased tumor growth, increased apoptosis, and reduced angiogenesis. The proposed mechanism was through reduced proliferating cell nuclear antigen and CD31 and increased caspase-3 and caspase-8.

The anticancer properties of ApoG2 have been investigated in the human B cell lymphoma line WSU-FSCCL. Cells treated with ApoG2 demonstrated antitumor effect in vivo through activation of caspase-3, caspase-8, and caspase-9 and increased cleavage of PARP and apoptosis-inducing factor [70].

Niu et al. [105] performed an in vivo study of mice bearing MCF-7 breast cancer cells. Treatment with ApoG2 suppressed colony formation in a time- and dose-dependent manner. Additionally, ApoG2 caused cell cycle arrest in the S and G2/M phases, triggered apoptosis, decreased Bcl-2 expression, and increased Bax and Beclin 1. When cells treated with ApoG2 are stained with acridine orange or observed in transmission electron microscope pictures, the autolysosomes become visible. In another study, mice bearing MCF-7 cells were treated with 40 μmol/L of ApoG2 for 72 h. ApoG2 increased apoptosis and autophagy. Zhang et al. [82] found that ApoG2 increased autophagy in PC-3 and LNCaP prostate cancer cells in vitro through the development of membrane vacuoles and the production of acidic vesicle organelles, which is a novel mechanistic finding. Furthermore, ApoG2 increased Beclin 1, microtubule-associated proteins 1A/1B light chain 3B, and 3-methyladenosine, an autophagy inhibitor, in both cell lines.

There has been at least one in vitro study regarding the use of ApoG2 in hepatocellular carcinoma (HCC). Cheng et al. [42] investigated the anticancer properties of ApoG2 in HepG2 and Hep3B HCC cells. ApoG2 induced autophagy through increased Beclin 1, resulting in inhibition of the mTOR pathway, ROS production, phosphorylation of extracellular signal-regulated kinase and JNK, and transfer of high mobility group box 1 from the nucleus to the cytoplasm. The addition of the antioxidant N-acetylcysteine reduced ROS-induced autophagy while increasing apoptosis and cell death, implying that the autophagy caused by ApoG2 is largely mediated through ROS. Therefore, combining ApoG2 with an antioxidant may be an effective chemotherapeutic regimen to improve drug sensitivity.

Mi et al. [58] investigated the anticancer propertied of ApoG2 when combined with the chemotherapy agent adriamycin for the treatment of SMMC-7721 HCC cells in vitro. The combination therapy induced cytotoxicity and apoptosis to a greater extent than when ApoG2 was used alone. The combined therapy increased apoptosis through downregulating Bcl-2, Mcl-1, and Bcl-xL, upregulating the proapoptotic protein Noxa, and increasing caspase-3 and caspase-9. Furthermore, there was increased p53 expression when ApoG2 was coupled with adriamycin after 48 h of therapy. The researchers also conducted an in vivo study using nude mice xenografted with SMMC-7721 cells. The mice were treated with 100 or 200 mg/kg of ApoG2 administered once daily for 28 days and 5.5 mg/kg of adriamycin given i.v. once a week for 4 weeks, which synergistically enhanced tumor suppression compared to ApoG2 alone [104].

It has been well established that the amount of copper ions present in malignant tumor tissues is significantly increased compared to healthy tissues. Zubair et al. [108] investigated the effects of both gossypol and ApoG2 in human peripheral lymphocytes. Both gossypol and its derivative induced oxidative damage and DNA fragmentation in these cells through the mobilization of endogenous copper ions. Compared to gossypol, ApoG2 demonstrated enhanced DNA breakage and increased ROS, suggesting that ApoG2 is more permeable in lymphocytes than gossypol. In a subsequent study by the same researchers, copper-mediated ROS generation and apoptosis driven by gossypol and ApoG2 in breast (MDA-MB-231), prostate (PC-3), and pancreatic (BxPC-3) cancer cells was investigated. Furthermore, they found that the sensitivity of MCF-10A breast epithelial cells to gossypol and ApoG2 can be potentiated by pretreatment with copper ions [109].

### 8.2. Gossypolone

A number of gossypolone derivatives have been synthesized to improve water solubility. 6-Aminopenicillanic acid sodium is one such gossypolone, which demonstrated improved water solubility and anticancer action in CT26 colon cancer cells in vitro. The researchers also utilized an in vivo model of CT26 tumor-bearing male BALB/c mice treated with 10 μmol/kg of 6-aminopenicillanic acid sodium-gossypolone daily through gavage for 12 days. 6 aminopenicillanic acid sodium-gossypolone downregulated Bcl-2 and Bcl-xL expression at the protein level, rather than the mRNA level, reduced toxicity, and exerted synergistic effect with 10 mg/kg of 5-fluorouracil, which was administered every 2 days [51].

Gossypolone and its ethylamine derivatives were evaluated against the human cervical cancer cell line KB in vitro. Cells treated with gossypolone and its derivatives demonstrated the greatest cytotoxicity, with median inhibitory concentration (IC50) values in the micromolar range, compared to gossypol, reduced gossypol, and Schiff bases of racemic extracts. In culture medium, there was improved cytotoxicity of gossypol and gossypolone in the absence of serum, while cytotoxicity was decreased in the presence of catalase and mannitol [59]. In another study, the cytotoxicity of chiral gossypol Schiff base and gossypolone was investigated in KB and MCF-7 cancer cells. However, the chiral gossypol Schiff base could not be sufficiently isolated due its instability. Gossypolone and other synthesized gossypol derivatives all induced cytotoxicity against adriblastina-resistant MCF/ADR cells, with gossylopolone demonstrating the greatest effects [50].

### 8.3. Gossypol Schiff Base

When treating cervical (HeLa), malignant glioma (U87), and gastric (M85) cancer cells, (+)-gossypol was transformed into gossypol Schiff base derivatives, which exhibited a much higher level of toxicity. Once this transformation has occurred, phenolic hydroxyl groups become the primary mediator of anticancer activity [110]. As an extension of the aforementioned study, Dao et al. [50] found that similar to gossypol, (−)-gossypol ethylamine is more toxic than (+)-gossypol ethylamine and racemic gossypol ethylamine to KB and MCF-7 cancer cells.

### 8.4. Miscellaneous Gossypol Derivatives

Novel gossypol and gossypolone dithiolane compounds have been isolated and found to be less hazardous to KB cells, however they were more harmful when used to create dehydrogossypoldithianes, dehydrogossypoldithiolanes, and gossypolone. In the presence of nitric oxide (NO), gossypol thiol derivatives became increasingly toxic, indicating that the dithiane or dithiolane gossypol derivatives can operate as antitumor prodrugs in tumor tissues with high NO concentrations [111]. Ultrasound-assisted interaction of gossypol potassium salt with 2,3,4,6-tetra-O-acetyl-α-D-glucopyranosyl bromide yielded unique compounds, including O-glucosidic gossypol isomers, 7,7′-gossypol diglucoside tetraacetate, 6,7′-gossypol diglucoside tetraacetate, 7,7′-gossypol diglycoside, and 6,7′-gossypol diglycoside. Additionally, O-glucosidic gossypol isomers demonstrated promising anticancer and antitrypanosomatid activities [11].

## 9. Synergistic Studies of Gossypol against Cancer Cells

When gossypol is utilized along with other anticancer agents, the combination is more effective than either drug alone [32]. Tomoda et al. [112] evaluated Pluronic P85, an ABA triblock copolymer of propylene oxide and ethylene oxide, against the human lung carcinoma cell line A549 in vitro and in vivo. (−)-Gossypol-loaded Pluronic P85 is a more powerful radio sensitizer in vitro and improved the antiproliferative activity of (−)-gossypol. In contrast, the combination of Pluronic P85 and (−)-gossypol substantially reduced clonogenic survival of A549 cells and increased radiation cancer cell death. In mice bearing A549 cancer cells, treatment with 200 mg/kg of Pluronic P85 (i.v.) daily and 15 mg/kg of (−)-gossypol (i.v.) daily for 5 days, increased radiation-related tumor control. When Pluronic P85 was administered alone, there was no reduction in tumor growth, compared to the suppression of tumor growth that occurred when the combination treatment was administered. In another study, valproic acid improved the antitumor effect of (−)-gossypol in vitro in Burkitt lymphoma Namalwa cells. Concentrations of 2 mmol/L of valproic acid and 5 μmol/L of (−)-gossypol suppressed cell proliferation to a greater degree when used together rather than when used individually [67].

Antiapoptotic genes, such as the Bcl-2 family, play a key role in conferring resistance to traditional chemotherapy. Gossypol has demonstrated the potential to overcome the drug resistance developed to certain medicines [65,113]. As a result, gossypol can enhance the induction of apoptosis when combined with chemotherapeutic medications. For example, an in vitro study by Cengiz et al. [80] treated PC-3 cancer cells with gossypol and docetaxel, which improved apoptosis. Similarly, combination of gossypol with gemcitabine overcame gemcitabine resistance in PC-3 cells with high Bcl-2 expression [114]. These findings suggest that the reversal of gemcitabine resistance seen with gossypol use involves downregulation of Bcl-2 and Bcl-xL and upregulation of Noxa and Mcl-1S. In a subsequent study, gossypol, paclitaxel, and cyclopamine mixed in poly (ethylene glycol)-blockpoly (ε-caprolactone) micelles were administered intraperitoneally to mice xenografted with ES-2 and SKOV3 ovarian cancer cells. The combination treatment suppressed tumor growth and prolonged survival in comparison to paclitaxel alone [115].

## 10. Clinical Studies

The antineoplastic potential of gossypol and its R-(−)-enantiomer (R-(−)-gossypol acetic acid, AT-101 have been investigated individually and in conjunction with traditional chemo-radiation regimens. A phase II clinical trial of AT-101 combined with other chemotherapy agents is currently underway in the United States [19]. Racemic gossypol, such as the compound gossypol acetate tablets made by Xi’an North Pharmaceutical Company, is already marketed as an antitumor medication in China. Low dosages of 30 mg daily or less of gossypol/AT-101 were found to be acceptable as monotherapy or in combination with chemoradiation when orally administered. If adverse reactions are experienced, dose adjustments can be made [116].

There have been four randomized controlled trials (RCTs) investigating the use of AT-101 in multiple cancer types (Table 5). Specifically, patients with NSCLC, HNSCC, gastroesophageal cancer, and castration-resistant prostate cancer were treated with varying doses of AT-101. The comparison standard of treatment was docetaxel, docetaxel plus cisplatin, or docetaxel plus prednisone. The greatest advantage was seen in high-risk patients, who experienced greater progression-free survival or improved survival [17].

Surprisingly, the most recent clinical trial coupled low-doses of AT-101 with docetaxel, fluorouracil, and radiation, which resulted in complete responses in high-risk patients and increased overall survival in 11 of 13 patients with gastroesophageal cancer, with a median time of progression of 12 months and progression-free survival for 52 months [117]. Since promising findings only occur in certain subgroups of patients [118], it is particularly important to identify which cancer types are the most sensitive to AT-101. More research is needed to conclude what the lowest recommended dose of gossypol is to maximize clinical efficacy and minimize toxicity. These findings should be verified in large placebo-controlled cohort RCTs.

## 11. Future Perspectives

While there are numerous studies that have been performed to evaluate the anticancer effects of gossypol, there remains limitations in the literature and room for future research. Despite the many mechanisms that were discussed in this review, more information should be obtained to understand all of the anticancer molecular targets of gossypol and their metabolites. Additional research on the synergistic effects of additional chemotherapeutic agents with gossypol and its derivatives is another area that needs to be explored.

More in vivo studies need to be performed, as the majority of current research is in vitro. A greater breadth of cancers should be included in the in vivo investigations, which should also confirm the in vitro mechanisms of action. The safety and toxicity of gossypol should be further investigated in vivo.

While there have been four clinical trials evaluating gossypol in the treatment of NSCLC, HNSCC, gastroesophageal cancer, and castration-resistant prostate cancer, there is a need for further clinical trials of gossypol and their derivatives against different cancer types. Furthermore, in order to optimize the bioavailability and anticancer effects of the active gossypol phytocompounds, additional studies are required to develop improved delivery mechanisms. Further research is required to establish the therapeutically effective dose of gossypol, the best combination with chemotherapeutic agents, and the high therapeutic value. Additionally, for other tumor entities or subtypes of cancer cells that may be impacted by gossypol treatment, the relationship between the therapeutic effect of gossypol and genetic modifications needs to be clinically studied.

## 12. Conclusions and Current Challenges/Limitations

Gossypol is a naturally-occurring yellow pigment that is widely present in cottonseeds. While it has been historically used as a male contraceptive agent, more recently the compound has been investigated for its promising anticancer potential. Several gossypol derivatives, including apogossypol, ApoG2, gossypolone, and gossypol Schiff base, have been isolated and evaluated for their anticancer effects. In this review, we provide a systematic analysis of the antineoplastic activity of gossypol and its derivatives, summarize the in vitro, in vivo, and clinical trials, and discuss combination therapies of gossypol with traditional chemotherapy agents. The sources, distribution, chemical structure, and toxicity of gossypol and its constituents are also briefly explored. More research is required to determine the precise mechanisms of gossypol’s absorption, interactions with the enzymes responsible for drug metabolism, and degradation, as well as the amount and routes of its excretion.

The current available pharmacokinetic studies of gossypol and its derivatives have shown that apogossypol has a slower clearance rate, broader AUC, and better microsomal stability than gossypol. Apogossypol exhibited identical oral and i.v. pharmacokinetic profiles, as well as in vitro stability. Furthermore, apogossypol hexaacetate transforms to apogossypol and lacks any quantitative oral bioavailability in vitro and in vivo.

This review highlights the therapeutic potential of gossypol in several cancer subtypes, including adrenal, breast, bladder, colon, head and neck, gastric, multiple myeloma, and prostate cancers. The anticancer effects of gossypol involve many molecular mechanisms, such as suppression of Bcl-2 family proteins, induction of caspase-dependent and mitochondrial pathway-mediated apoptosis, modulatory effects on the cell cycle and cell signaling pathways, autophagy, generation of ROS, and interference with tyrosine receptor protein kinase. These anticancer pathways and mechanisms are summarized in Figure 7 and Figure 8.

There have been 55 in vitro studies and 5 in vivo studies regarding the antineoplastic effects of gossypol and their derivatives against various cancer types. There have been four RCTs investigating the use of AT-101 in multiple cancer types. Additionally, phase I and II clinical trials have evaluated AT-101 when combined with other chemotherapy agents. Gossypol acts synergistically with traditional cancer medications to improve susceptibility of cancers to treatment. Many studies have found that drug resistant cancer cells are sensitive to gossypol and its derivatives.

Taking into consideration the in-depth analysis of current research as presented in this review, gossypol-derived bioactive phytochemicals possess significant potential for cancer therapy in many cancer types. Gossypol’s biological activity, particularly its apoptotic and antiproliferative properties, have been widely studied. Nevertheless, further research should be gathered to fully understand the chemotherapeutic potential of gossypol.

## Figures and Tables

**Figure 1 pharmaceutics-14-02624-f001:**
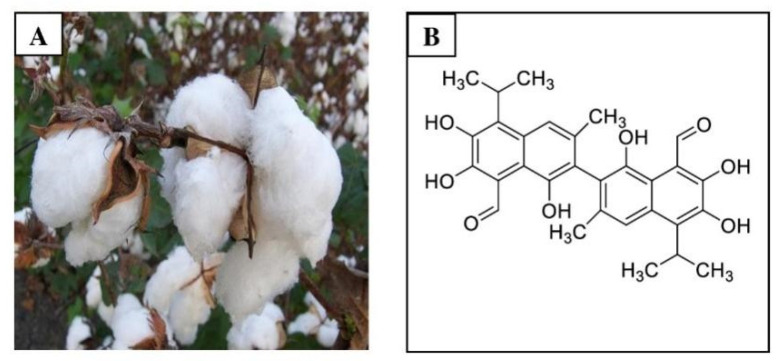
Gossypol profile: (**A**) cotton plant (*Gossypium hirsutum* L.) (**B**) structure of gossypol.

**Figure 2 pharmaceutics-14-02624-f002:**
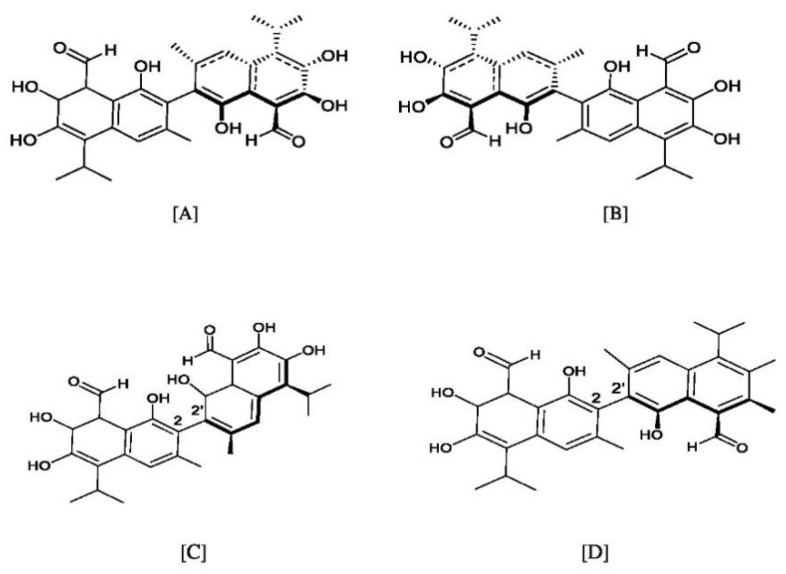
Structures of the gossypol enantiomers: (**A**) and (**B**), absolute configuration of gossypol atropisomers; (**C**) (*aR*)-gossypol (M) (−)-1; and (**D**) (*aS*)-gossypol (P) (+)-1.

**Figure 3 pharmaceutics-14-02624-f003:**
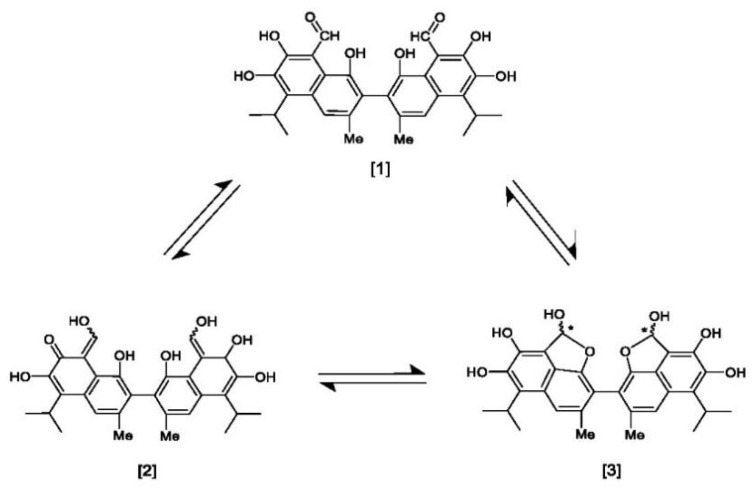
Symmetrical tautomeric forms of gossypol: [1] dialdehyde, [2] diketone, and [3] dilactol (* indicates a new asymmetric center in the molecule).

**Figure 4 pharmaceutics-14-02624-f004:**
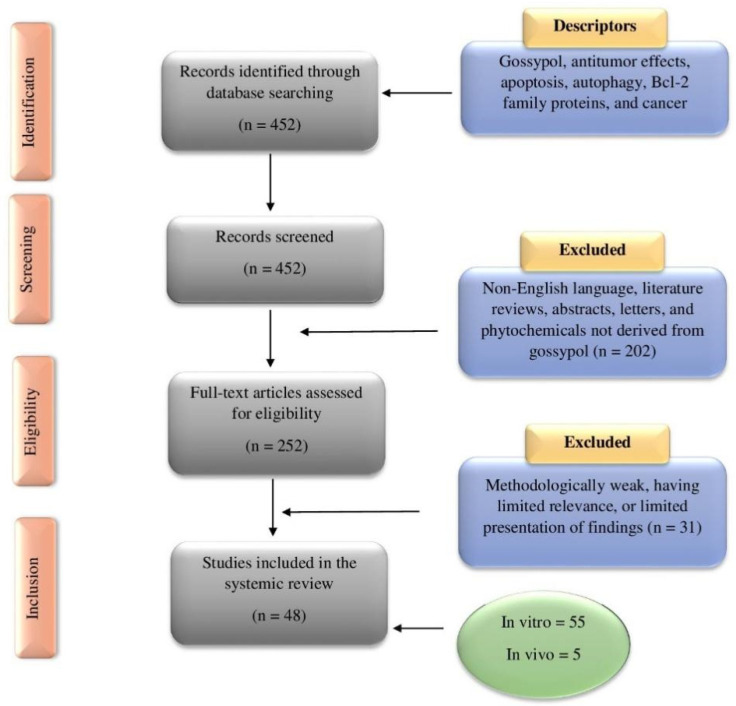
A PRISMA flowchart illustrating the literature search and study selection process relevant to the anticancer potential of gossypol. The total number of in vitro and in vivo studies [42] is greater than the number of studies included in this work [43] because many publications contained results from more than one organ-specific cancer or study type (in vitro or in vivo).

**Figure 5 pharmaceutics-14-02624-f005:**
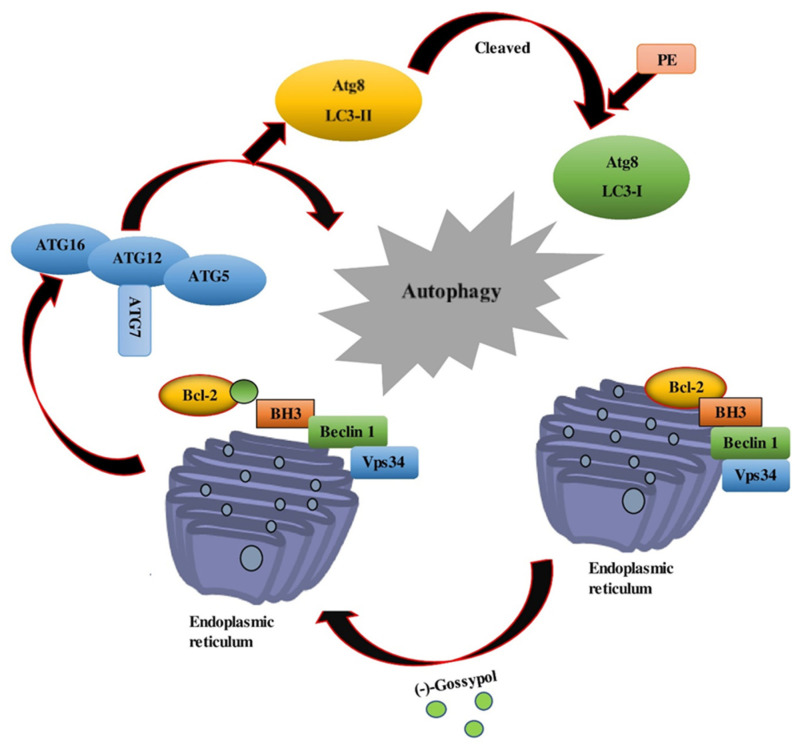
(−)-Gossypol was found to be more effective at inducing autophagy in apoptosis-resistant cancer cells with high Bcl-2 and Bcl-xL levels.

**Figure 6 pharmaceutics-14-02624-f006:**
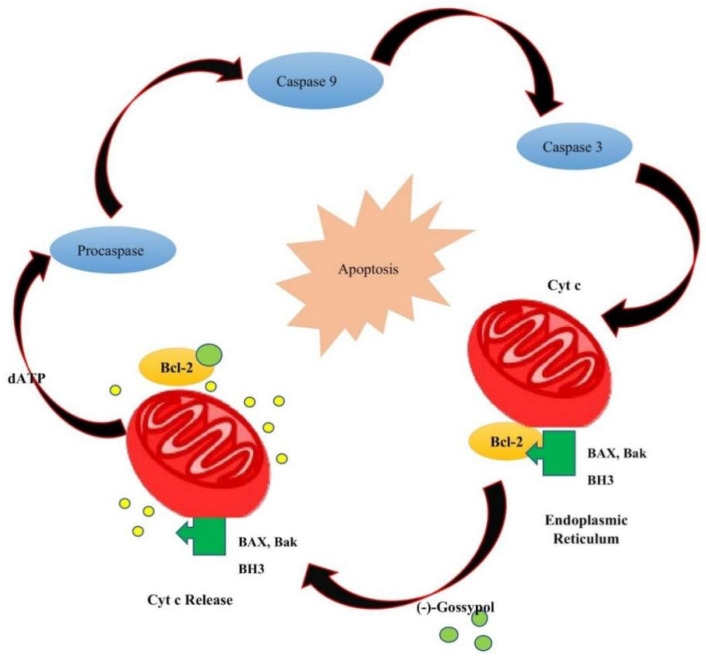
In apoptosis-sensitive cancer cells with low levels of antiapoptotic proteins Bcl-2 and Bcl-xL, (−)-gossypol is known to trigger apoptosis.

**Figure 7 pharmaceutics-14-02624-f007:**
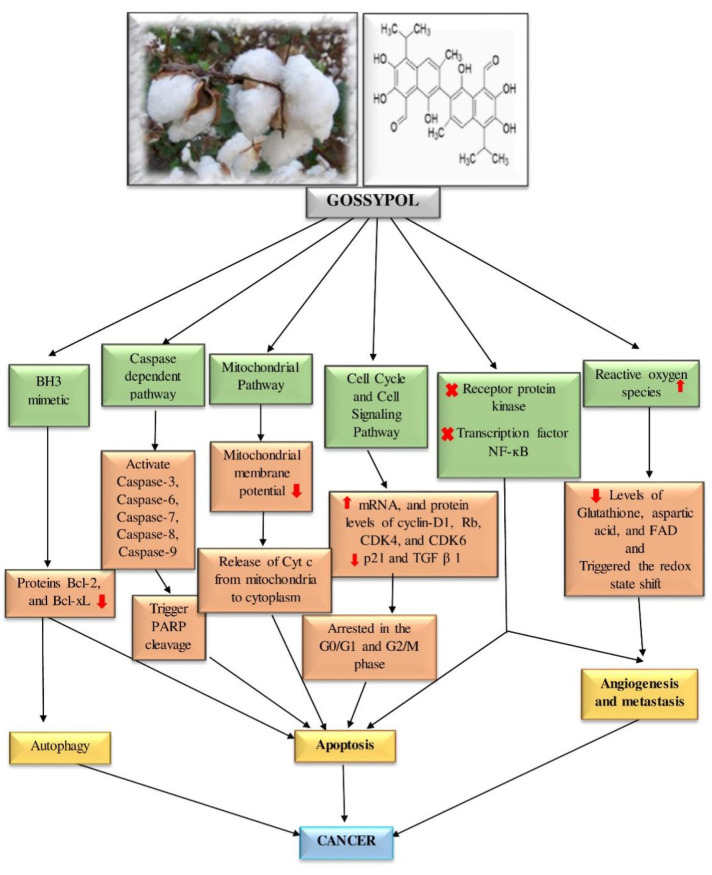
Gossypol’s activity against cancerous cells by affecting different cellular pathways and enzymes. ↑, increase or upregulation; ↓ decrease or downregulation; x, inhibition.

**Figure 8 pharmaceutics-14-02624-f008:**
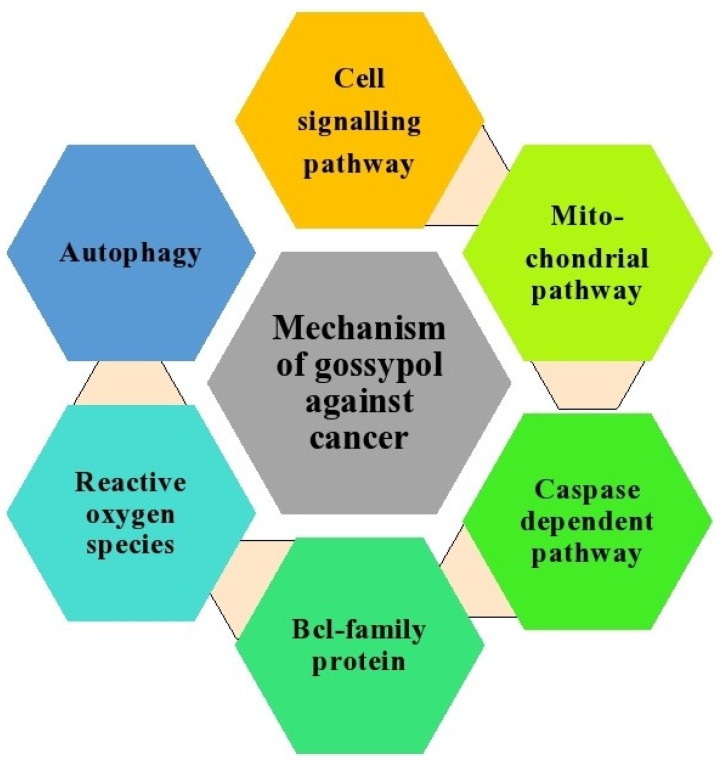
The molecular mechanism of actions of gossypol and its derivatives as anticancer agents.

**Table 1 pharmaceutics-14-02624-t001:** Plant species containing gossypol [9].

Plant Species	Location of Gossypol in the Plant	Gossypol Enantiomer in Excess
*Gossypium hirsutum*	Seeds and roots	(+)
*Gossypium barbadense*	Seeds	(−)
*Gossypium aboreum*	Seeds, stem and roots	(+)
*Gossypium herbaceum*	Seeds and stem	(+)
*Gossypium mustelinum*	Seeds	(+)
*Thespesia populnea*	Wood, leaves and flowers	(+)

**Table 2 pharmaceutics-14-02624-t002:** Various gossypol derivatives and their structures [17,18].

Gossypol Derivatives	Structure
Gossypol	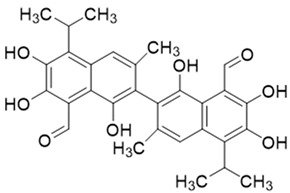
Apogossypolone	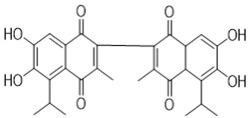
Apogossypol	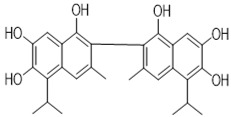
Gossypolone	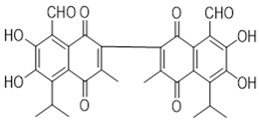
6-Aminopenicillanic acid sodium gossypolone	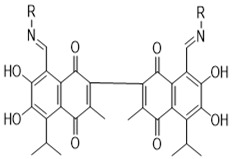
BI-97C1	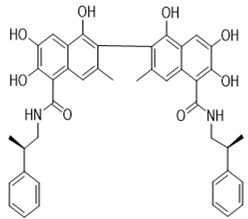

**Table 3 pharmaceutics-14-02624-t003:** Gossypol and their derivative’s activities against different types of cancer cell lines in vitro.

Compound Tested	Cell Lines Used	IC_50_ (μM) (Duration)	References
*Adrenal cancer*			
Gossypol	SW-13, H295r (human adrenocortical carcinoma)	1.3–2.9	[46]
Apogossypol hexaacetate	H295r, SW-13 (human adrenocortical carcinoma)	5.2–6.8	[46]
*Bladder cancer*			
Gossypol	UM-UC2 and UM-UC-9 (human bladder carcinoma)	0–20	[47]
*Breast cancer*			
Gossypol	MDA-MD-231 (human breast carcinoma)	14.37	[11]
Gossypol	MCF-7, MDA-MD-231 (human breast carcinoma)	3.99–4.64 (72 h)	[7]
Gossypol	MCF-7, MCF-6/ADR, MDA-MD-231 (human breast carcinoma)	None reported	[48]
Gossypol	MCF-7 (human breast carcinoma)	None reported	[49]
ApoG2	MCF-7 (human breast carcinoma)	None reported	[39]
Gossypolone	MCF-7, MCF-6/ADR, MDA-MD-231 (human breast carcinoma)	None reported	[43]
Gossypolone	MCF-7/ADR, MCF-7/WT (human breast carcinoma)	0.28–0.36	[50]
6-Aminopenicillanic acid sodium gossypolone	MCF-7, MDA-MB435 (human breast carcinoma)	29.06–40	[51]
6-Aminopenicillanic acid sodium gossypolone	4T1 (mouse mammary carcinoma)	28.25	[51]
Gossypol-enriched cottonseed oil	MDA-MD-231, MDA-MD-468 (human breast carcinoma)	21.85–26.39	[52]
Ethylimine of (±)-gossypol	MCF-7/ADR, MCF-7/WT (human breast carcinoma)	0.40–0.45	[50]
Ethylimine of (±)- gossypolone	MCF-7/ADR, MCF-7/WT (human breast carcinoma)	0.29–0.35	[50]
Methylimine of (±)-gossypolone	MCF-7/ADR, MCF-7/WT (human breast carcinoma)	0.14–0.37	[50]
*Gastrointestinal and associated cancers*			
Gossypol	HCT116 (human colorectal carcinoma)	8.8	[53]
Gossypol	COLO 225 (human colon carcinoma)	0.1–100	[8]
Gossypol	HT-29 (human colorectal carcinoma)	10	[54]
6-Aminopenicillanic acid sodium gossypolone	HCT116, SW620 (human colorectal carcinoma)	6.56–17.35	[51]
6-Aminopenicillanic acid sodium gossypolone	CT26 (mouse colon carcinoma)	6.59	[51]
Gossypol	M85 (human gastric carcinoma)	18.4–39.7	[55]
ApoG2	MKN28, MKN45, AGS (human gastric carcinoma)	5.6–11.37 (72 h)	[56]
ApoG2	SGC-7901 (human gastric carcinoma)	18.7	[57]
ApoG2	SMMC-7721 (human hepatocellular carcinoma)	17.29–30.63	[58]
ApoG2	HepG2, Hep3B (human hepatocellular carcinoma)	None reported	[42]
*Gynecologic cancers*			
Gossypol	SKOV-3 (human ovarian carcinoma)	5.7	[46]
Gossypol	KB (human cervical carcinoma)	5.7	[59]
Gossypol	RL95-2 (human endometrial carcinoma)	3.4	[46]
Gossypol	HeLa (human cervical carcinoma)	17.8–31.3	[55]
Apogossypol hexaacetate	Skov-3 (human ovarian carcinoma)	9.0	[46]
Apogossypol hexaacetate	RL95-2 (human endometrial carcinoma)	7.3	[46]
Gossypolone	KB (human cervical carcinoma)	0.45	[50]
Gossypolone	KB (human cervical carcinoma)	2.4	[59]
Ethylimine of (±)-gossypol	KB (human cervical carcinoma)	1.00	[50]
Ethylimine of (±)- gossypolone	KB (human cervical carcinoma)	0.55	[50]
Methylimine of (±)-gossypolone	KB (human cervical carcinoma)	0.50	[50]
*Head and neck cancers*			
Gossypol	Um-SCC-1, Um-SCC-17b, UM-SCC-5PT, UM-SCC-10BPT (human HNSCC)	3–15	[60]
Gossypol	UM-SCC-12, UM-SCC-23, UM-SCC-1, UM-SCC-6, UM-SCC-14A, UM-SCC-74B, UM-SCC-81B, UM-SCC-22A, UM-SCC-17B, UM-SCC-25 (human HNSCC)	3.75–6.85	[61]
ApoG2	C666-1, CNE-1, CNE-2, HONE-1 (human nasopharyngeal carcinoma)	0.908–66.01	[62]
ApoG2	CNE-2, HONE-1 (human nasopharyngeal carcinoma)	None reported	[63]
*Hematologic cancers*			
Gossypol	HL60 (human acute myeloid leukemia)	2.0–8.1 (24–48 h)	[64]
Gossypol	Vector/Jurkat, Bcl-2/Jurkat, Bcl-xL/Jurkat (human T cell leukemia)	4.3–26.6	[65]
Gossypol	U266, ARK, MM1, H929, OC1-MY5, OPM2, ARP-1, KMS11, CAG (human multiple myeloma)	< 6	[66]
Gossypol	Namalwa (human Burkitt lymphoma)	None reported	[67]
Gossypol	Namalwa (human Burkitt lymphoma)	None reported	[68]
Gossypol	KAS-6/1, MM1.S, OPM-1, OPM-2, U266 (human multiple myeloma)	2.5–7.5	[69]
ApoG2	WSU-FSCCL (human B cell lymphoma)	0.109	[70]
*Lung cancer*			
Gossypol	H1975 (human lung adenocarcinoma)	10.05–11.73	[71]
Gossypol	H1975 (human lung adenocarcinoma)	34.17–36.35	[72]
6-Aminopenicillanic acid sodium gossypolone	A549 (human lung adenocarcinoma)	8.77	[51]
BI-97C1	H460 (human large cell lung carcinoma)	0.33–0.51	[73]
*Neural cancers*			
Gossypol	U87 (human malignant glioma)	30.2–59.6	[55]
*Prostate cancer*			
Gossypol	PC-3 (human prostate carcinoma)	9.096	[11]
Gossypol	PC-3 (human prostate carcinoma)	4.74	[74]
Gossypol	PC-3, LNCaP (human prostate carcinoma)	2.8–3.5	[75]
Gossypol	MAT-lylu (rat prostate carcinoma)	14.5	[76]
Gossypol	MAT-lylu (rat prostate carcinoma)	None reported	[77]
Gossypol	PC-3 (human prostate carcinoma)	None reported	[78]
Gossypol	DU-145 (human prostate carcinoma)	None reported	[79]
Gossypol	PC-3 (human prostate carcinoma)	10 (24–72 h)	[80]
Gossypol	PC-3 (human prostate carcinoma)	None reported	[81]
ApoG2	PC-3, LNCaP (human prostate carcinoma)	None reported	[82]
BI-97C1	PC-3 (human prostate carcinoma)	0.11–0.15	[73]
6-Aminopenicillanic acid sodium gossypolone	PC-3 (human prostate carcinoma)	29.46	[51]
*Skin cancer*			
6-Aminopenicillanic acid sodium gossypolone	B16-F10 (mouse melanoma)	11.05	[51]
*Thyroid cancer*			
Apogossypol hexaacetate	TT (human thyroid carcinoma)	18.9	[46]

**Table 4 pharmaceutics-14-02624-t004:** Gossypol and their derivative’s activities against different types of in vivo cancer models.

Compound Tested	Animal Tumor Models	Anticancer Effects	Mechanisms	Dose (Route)	Duration	References
*Breast cancer*						
Apogossypolone	Athymic nude female mice xenografted with MCF-7 breast carcinoma cells	Inhibited tumor growth	↑Apoptosis; ↑autophagy	120 mg/kg/day (i.p.)	4 weeks	[105]
*Gastrointestinal and associated cancers*						
Combination of apogossypolone and adriamycin	BALB/c *nu*/*nu* mice xenografted with SMMC-7721 hepatocellular carcinoma cells	Inhibited tumor growth	↑Apoptosis; ↑DNA fragmentation	100 or 200 mg/kg/day (intragastric) ± 5.5 mg/kg/week adriamycin (i.v.)	28 days	[58]
*Head and neck cancers*						
Apogossypolone	Athymic nude (*nu*/*nu*) mice xenografted with CNE-1 or CNE-2 nasopharyngeal carcinoma cells	Inhibited tumor growth and angiogenesis; enhanced antitumor activity of cisplatin (CNE-2 cells only)	↑Apoptosis; ↓CD31	200 mg/kg/day (intragastric) ± 3 mg/kg; every 2 days (i.p.)cisplatin		[62]
*Hematologic cancers*						
ApoG2	Female ICR SCID mice xenografted with WSU-FSCCL B cell lymphoma cells	Inhibited tumor growth	↑Caspase-3; ↑caspase-8; ↑caspase-9; ↑PARP; ↑AIF	25 mg/kg/day (i.p. or i.v.)	5 days	[70]
*Prostate cancer*						
(−)-Gossypol	Male BALB/c nude mice xenografted with PC-3 prostate carcinoma cells	Inhibited tumor growth and angiogenesis	↓VEGF	15 mg/kg/day (intratumoral)	50 days	[106]
(−)-Gossypol	Male BALB/c nude mice xenografted with PC-3 prostate carcinoma cells	Inhibited tumor growth	↓Bcl-2; ↑caspase-3; ↑caspase-8; ↓CD31; ↓PCNA	2.5–10 mg/kg/day (i.p.)	7 days	[74]
AT-101(R-(–)-gossypol acetic acid	SCID mice xenografted with VCaP prostate carcinoma cells	Inhibited tumor growth	↑Apoptosis; ↑Bcl-2; ↑androgen receptor expression	15 mg/kg; 5 days/week (p.o.)	6 weeks	[104]

Symbols and abbreviations: ↑, increase or upregulation; ↓ decrease or downregulation; AIF, apoptosis inducing factor; ApoG2, apogossypolone; i.p., intraperitoneal; i.v, intravenous; PARP, poly (ADP-ribose) polymerase; PCNA, proliferating cell nuclear antigen; p.o., per os; VEGF, vascular endothelial growth factor.

**Table 5 pharmaceutics-14-02624-t005:** Oral application of single-agent gossypol/AT-101 investigated in cancer patients in Phase I and II trials.

Clinical Trial (Date)	Trial Design	Cancer Type (Number of Patients)	Dose, Route, Frequency (Duration)	Adverse Events	Primary Outcome	References
NCT00848016 (2019)	Phase II	Adrenocortical carcinoma (*n* = 29)	20 mg AT-101 PO QD (21 days/month × 80 cycles)	Elevated cardiac troponin, hypokalemia, GI upset, elevated AST/ALT and fatigue	PD: 27; PR: 0; SD: 8 Median duration of 3.8 months; median time of progression 1.9 months; mOS: 8.5 months	[119]
NCT00773955 (2011)	Phase II	Small cell lung carcinoma (*n* = 14)	20 mg AT-101 PO QD (21 days/month × 6 cycles)	GI upset, exhaustion, anorexia and hematologic abnormalities	OR: 0; SD: 3 Median time of progression 1.7 months; mOS: 8.5 months	[120]
NCT00286806 (2009)	Phase I/II	Castrate-resistant prostate carcinoma (*n* = 23)	30 mg AT-101 (21 days/month) chemotherapy naive patients, ≥eight weeks of therapy	GI upset, elevated AST/ALT and small intestinal obstruction	Modest single-agent activity; treatment was generally well tolerated	[118]
(2001)	Phase I/II	Breast carcinoma (*n* = 20)	30–50 mg AT-101 QD; Pre-treatment with doxorubicin and paclitaxel	GI upset, fatigue and dysgeusia	MR: 1; SD: 2 Blood gossypol levels are 10-fold lower than in vitro levels; No clear correlation between plasma drug levels and gossypol dose	[40]
(1999)	Phase II	Radiation-resistant glial carcinoma (*n* = 27)	10 mg racemic gossypol acetic acid p.o. QD or BID	Mild toxicity	PR: 2; SD: 4; PD: 21 No difference in plasma levels between responders and non-responders	[121]
(1993)	Phase I	Adrenal carcinoma (*n* = 21)	30–70 mg racemic gossypol mitotane, and suramine	No serious adverse events	PR: 3 (50% reduction in tumor volume); MR: 1; PD: 13 No significant decrease in steroid secretion	[122]
(1992)	Phase I	Human carcinoma (*n* = 34)	30–180 mg gossypol acetic acid; followed by 30 mg twice weekly, QD, or BID	No serious adverse events	SD: 3; PD: 20 No association between serum drug levels and gossypol dose	[123]

Abbreviations: ALT, alanine aminotransferase; AST, aspartate aminotransferase; BID, twice daily; GI, gastrointestinal; mOS, median overall survival; MR, minor response; n, number of subjects; OR, objective response; PD, progressive disease; p.o., per os; PR, partial response; QD, daily; SD, stable disease.

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
