# Peer review of "Gossypol and Its Natural Derivatives: Multitargeted Phytochemicals as Potential Drug Candidates for Oncologic Diseases"

_pharmaceutics, 2022, doi:10.3390/pharmaceutics14122624_

Round 1

Reviewer 1 Report

Comments to the authors:

The manuscript is well written and designed. However, I have a problem with the novelty of the manuscript. Just recently, a review paper has been published dealing with gossypol in cancer clinical trials (Renner et al. Pharmaceuticals. 2022;15:144. doi: 10.3390/ph15020144). The references provided in the present manuscript are identical to the ones in the published article (please note: references in table 5 seem not to match with the reference list). Interestingly, the respective manuscript has been cited (reference 16) indicating that the authors have been aware of this issue. Therefore, it is not fair to leave this information out, just mention that "gossypol has been investigated for its anticancer activity" followed by "no clinical studies were included". The last part has been related to reference 15, still, the readership will be driven to assume that no clinical studies have been reported at all. Secondly, of 131 articles cited, only 9 are from 2020 and newer, and of these, only one article is related to gossypol's mode of action. Many articles have been published since 2020 dealing with gossypol and cancer. I recommend to take care on these articles. Finally, chapter 3 ("Chemistry of Gossypol") has already been presented in Renner et al. (as well as others) and should be shortened.

Author Response

The authors of this manuscript express their sincere thanks to the reviewer for the critical assessment of this work. The authors have acted upon the recommendations of the reviewer which have resulted in a significant enhancement in the quality of this manuscript. All modifications incorporated in the manuscript are highlighted in a red color font. A “point-by-point” response to each comment is outlined below.

Comment 1:

The manuscript is well written and designed. However, I have a problem with the novelty of the manuscript. Just recently, a review paper has been published dealing with gossypol in cancer clinical trials (Renner et al. Pharmaceuticals. 2022;15:144. doi: 10.3390/ph15020144). The references provided in the present manuscript are identical to the ones in the published article (please note: references in table 5 seem not to match with the reference list). Interestingly, the respective manuscript has been cited (reference 16) indicating that the authors have been aware of this issue. Therefore, it is not fair to leave this information out, just mention that "gossypol has been investigated for its anticancer activity" followed by "no clinical studies were included". The last part has been related to reference 15, still, the readership will be driven to assume that no clinical studies have been reported at all. Secondly, of 131 articles cited, only 9 are from 2020 and newer, and of these, only one article is related to gossypol's mode of action. Many articles have been published since 2020 dealing with gossypol and cancer. I recommend to take care on these articles.

Response:

We sincerely appreciate the valuable suggestions which we have found extremely useful while revising our manuscript. We understand that Renner and colleagues review clinical trials involving gossypol; however, the novelty of our paper involves summarizing in vitro and in vivo studies in an organ system-specific manner while detailing the molecular mechanisms of action. The references involving Table 5 and the clinical studies section of the text have also been revised (pages 23-25). We have also removed the portion of the text stating “no clinical studies were included” to improve the clarity of this manuscript.    

Comment 2:

Finally, chapter 3 ("Chemistry of Gossypol") has already been presented in Renner et al. (as well as others) and should be shortened.

Response:

We thank the reviewer for their feedback and have removed multiple sentences from Section 3 as suggested (page 4, lines 118-200; page 5, line 128). 

Additionally,

  1. The entire manuscript has been thoroughly checked and edited to minimize typographical errors as well as to ensure uniform style, organization, and quality.
  2. The reference list has been modified as we have added several new references. Special attention is given to conform to the order of references and bibliographic style of the journal.

Finally,

On behalf of my co-authors, I once again express my sincere thanks to the erudite reviewer for the valuable suggestions and constructive input to improve the quality of our manuscript.

Reviewer 2 Report

Although there are many studies in the specialized literature who evaluates the anticancer effects of gossypol and its derivative the review “Gossypol and Its Natural Derivatives: Multitargeted Phytochemicals as Potential Drug Candidates for Oncologic Diseases” describe recent advances in the chemotherapeutic potential of gossypol and related compounds against various malignancies and in combination with other chemotherapeutic agents,  identify limitations in the current literature, and suggest future research directions. The article is well written, it clearly presents the cited articles and how they were selected, however, some corrections should be made in order to be published. You can find these suggestions in the attached document.

Author Response

The authors of this manuscript express their sincere thanks to the reviewer for the critical assessment of this work. The authors have acted upon the recommendations of the reviewer which have resulted in a significant enhancement in the quality of this manuscript. All modifications incorporated in the manuscript are highlighted in a red color font. A “point-by-point” response to each comment is outlined below.

General comments:

Although there are many studies in the specialized literature who evaluates the anticancer effects of gossypol and its derivative the review “Gossypol and Its Natural Derivatives: Multitargeted Phytochemicals as Potential Drug Candidates for Oncologic Diseases” describe recent advances in the chemotherapeutic potential of gossypol and related compounds against various malignancies and in combination with other chemotherapeutic agents,  identify limitations in the current literature, and suggest future research directions. The article is well written, it clearly presents the cited articles and how they were selected, however, some corrections should be made in order to be published. You can find these suggestions in the attached document.

Response:

We thank the reviewer for their expertise, time, and effort for reviewing our manuscript. We are deeply encouraged by the generous comments regarding the quality of our work. We sincerely appreciate the specific recommendations which we have found extremely valuable while revising our manuscript.

Specific comments:

Comment 1:

Row 57 and 62 and throughout the text - in vitro in vivo- should be written in italic

Response:

We appreciate the reviewer’s comment and have italicized the terms in vitro and in vivo throughout the entirety of the text.

Comment 2:

142 - 4. Toxicity of Gossypol

In this section toxicity is discussed such as genotoxic potential, gastrointestinal toxicity, maybe some concentrations could be mentioned or maximum tolerated dose. There are data in the literature regarding the concentrations at which gossypol and derivatives it starts to be toxic?

What about hepatotoxicity?

Response:

We thank the reviewer for their question and have thoroughly reviewed the literature to address this. To the best of our knowledge, there have been no studies performed regarding the concentrations at which gossypol exhibits toxicity. There has also been no mention of hepatotoxicity in the literature.

Comment 3:

241 -259 It is not very clear for me if the studies of Macoska et al. [56] took place in vitro or in vivo. The last statement refers to the effect of -gossypol as an adjuvant of chemotherapy may improve response rates in patients with advanced bladder cancers and in vitro study is mentioned on the line 242.

Response:

We thank the reviewer for their close attention to detail and have corrected the text to reflect that Macoska et al. is an in vitro study (page 12, line 252).

Comment 4:

For some types of cancer / cancer cells, the description of the effect obtained by the author is very brief for example in adrenal cancer, lung cancer ….are there few data provided by the authors in the cited studies?

Response:

We appreciate the reviewer’s question. It is the author’s opinions that brevity is best when discussing the individual in vitro and in vivo studies. Throughout the manuscript, we have summarized the findings from the studies to a few sentences each.    

Comment 5:

Also I recommend checking the spelling and guidelines of the journal.

Response:

We are grateful for the suggestions. We have thoroughly checked and edited our manuscript. The journal guidelines have been followed.

Additionally,

  1. The entire manuscript has been thoroughly checked and edited to minimize typographical errors as well as to ensure uniform style, organization, and quality.
  2. The reference list has been modified as we have added several new references. Special attention is given to conform to the order of references and bibliographic style of the journal.

Finally,

On behalf of my co-authors, I once again express my sincere thanks to the erudite reviewer for the valuable suggestions and constructive input to improve the quality of our manuscript.

Reviewer 3 Report

1. This review is very comprehensive. Only a few issues detected in the text should be addressed (please, see pdf).

2. Table formatting needs to be modified, and the picture may be modified to be more beautiful

Author Response

The authors of this manuscript express their sincere thanks to the reviewer for the critical assessment of this work. The authors have acted upon the recommendations of the reviewer which have resulted in a significant enhancement in the quality of this manuscript. All modifications incorporated in the manuscript are highlighted in a red color font. A “point-by-point” response to each comment is outlined below.

Comment 1:

This review is very comprehensive. Only a few issues detected in the text should be addressed (please, see pdf).

Response:

We thank the reviewer for his/her expertise, time, and effort for reviewing our manuscript and sincerely appreciate the valuable suggestions, which we have found extremely useful while revising our manuscript.

Comment 2:

Table formatting needs to be modified, and the picture may be modified to be more beautiful.

Response:

We thank the reviewer for the comment. We have revised several tables. We have tried our best to present various relevant key information in the figures.

Moreover, we are indebted to the reviewer for providing an annotated version of our manuscript with editing which we have found to be extremely helpful. We have executed the corrections suggested. All changed are marked with red color font.

Additionally,

  1. The entire manuscript has been thoroughly checked and edited to minimize typographical errors as well as to ensure uniform style, organization, and quality.
  2. The reference list has been modified as we have added several new references. Special attention is given to conform to the order of references and bibliographic style of the journal.

Finally,

On behalf of my co-authors, I once again express my sincere thanks to the erudite reviewer for the valuable suggestions and constructive input to improve the quality of our manuscript.

Round 2

Reviewer 2 Report

I believe that the article can be published in this form.

Author Response

Comment:

I believe that the article can be published in this form.

Response:

Thank you for checking our revised manuscript. We are pleased to learn that our manuscript can be published in its present form.